# PAC-NeRF: Physics Augmented Continuum Neural Radiance Fields for Geometry-Agnostic System Identification

**Xuan Li[1,*], Yi-Ling Qiao[2], Peter Yichen Chen[3,4], Krishna Murthy Jatavallabhula[3], Ming Lin[2], Chenfanfu Jiang[1], Chuang Gan[5,6]**
[1]UC Los Angeles, [2]University of Maryland, [3]MIT CSAIL, [4]Columbia University,
[5]UMass Amherst, [6]MIT-IBM Watson AI Lab
xuanli1@math.ucla.edu, yilingq@umd.edu, pyc@csail.mit.edu,
jkrishna@mit.edu, lin@cs.unc.edu, cffjiang@math.ucla.edu,
chuangg@umass.edu

## Abstract

Existing approaches to system identification (estimating the physical parameters of an object) from videos assume known object geometries. This precludes their applicability in a vast majority of scenes where object geometries are complex or unknown. In this work, we aim to identify parameters characterizing a physical system from a set of multi-view videos *without any assumption on object geometry or topology*. To this end, we propose "Physics Augmented Continuum Neural Radiance Fields" (PAC-NeRF), to estimate both the unknown geometry and physical parameters of highly dynamic objects from multi-view videos. We design PAC-NeRF to only ever produce physically plausible states by enforcing the neural radiance field to follow the conservation laws of continuum mechanics. For this, we design a hybrid Eulerian-Lagrangian representation of the neural radiance field, i.e., we use the Eulerian grid representation for NeRF density and color fields, while advecting the neural radiance fields via Lagrangian particles. This hybrid Eulerian-Lagrangian representation seamlessly blends efficient neural rendering with the material point method (MPM) for robust differentiable physics simulation. We validate the effectiveness of our proposed framework on geometry and physical parameter estimation over a vast range of materials, including elastic bodies, plasticine, sand, Newtonian and non-Newtonian fluids, and demonstrate significant performance gain on most tasks[1].

## 1 Introduction

Inferring the geometric and physical properties of an object directly from visual observations is a long-standing challenge in computer vision and artificial intelligence. Current machine vision systems are unable to disentangle the geometric structure of the scene, the dynamics of moving objects, and the mechanisms underlying the imaging process – an innate cognitive process in human perception. For example, by merely watching someone kneading and rolling dough, we are able to disentangle the dough from background clutter, form a predictive model of its dynamics, and estimate physical properties, such as its consistency to be able to replicate the recipe. There exists a large body of work on inferring the geometric (*extrinsic*) structure of the world from multiple images (e.g., structure-from-motion (Hartley & Zisserman, 2003)). This has been bolstered by recent approaches leveraging differentiable rendering pipelines (Tewari et al., 2022) and neural scene representations (Xie et al., 2022), unlocking a new level of performance and visual realism. On the other hand, approaches to extract the physical (*intrinsic*) properties (e.g., mass, friction, viscosity) from images are yet nascent (Jatavallabhula et al., 2020; Ma et al., 2021; Jaques et al., 2022; 2020) – all assume full knowledge of the geometric structure of the scene, thereby limiting their applicability.

The key question we ask in this work is "*can we recover both the geometric structure and the physical properties of a wide range of objects from multi-view video sequences*"? This dispenses with all

---

*This work was done during an internship at the MIT-IBM Watson AI Lab

[1]Demos are available on the project webpage: https://sites.google.com/view/PAC-NeRF

of the assumptions made by state-of-the-art approaches to video-based system identification (known geometries in Ma et al. (2021) and additionally rendering configurations in Jatavallabhula et al. (2020)). Additionally, the best performing approaches to recover geometries (but not physical properties) of dynamic objects in videos include variants of neural radiance fields (NeRF) (Mildenhall et al., 2020), such as (Pumarola et al., 2021; Tretschk et al., 2021; Park et al., 2021). However, all such neural representations of dynamic scenes need to learn object dynamics from scratch, requiring a significant amount of data to do so, while also being uninterpretable. We instead employ a differentiable physics simulator as a more prescriptive, data-efficient, and generalizable dynamics model; enabling parameter estimation solely from videos.

Our approach—Physics Augmented Continuum Neural Radiance Fields (PAC-NeRF)—is a novel system identification technique that assumes nothing about the geometric structure of a system. PAC-NeRF is extremely general – operating on deformable solids, granular media, plastics, and Newtonian/non-Newtonian fluids. PAC-NeRF brings together the best of both worlds; differentiable physics and neural radiance fields for dynamic scenes. By augmenting a NeRF with a differentiable continuum dynamics model, we obtain a unified model that estimates object geometries *and* their physical properties in a single framework.

Specifically, a PAC-NeRF $\mathcal{F}$ is a NeRF, comprising a volume density field and a color field, coupled with a velocity field $\mathbf{v}$ that admits the continuum conservation law: $\frac{D\mathcal{F}}{Dt} = 0$ (Spencer, 2004). In conjunction with a hybrid Eulerian-Lagrangian formulation, this allows us to advect geometry and appearance attributes to all frames in a video sequence, enabling the specification of a reconstruction error in the image space. This error term is minimized by gradient-based optimization, leveraging the differentiability of the entire computation graph, and enables system identification over a wide range of physical systems, where neither the geometry nor the rendering configurations are known. Our hybrid representation considerably speeds up the original MLP-based NeRF by efficient voxel discretization Sun et al. (2022), and also conveniently handles collisions in continuum simulations, following the MPM pipeline Jiang et al. (2015). The joint differentiable rendering-simulation pipeline with a unified Eulerian-Lagrangian conversion is highly optimized for high-performance computing on GPU.

In summary, we make the following contributions.

- We propose **PAC-NeRF** – a dynamic neural radiance field that satisfies the continuum conservation law (Section 3.1).

- We introduce a *hybrid Eulerian-Lagrangian representation*, seamlessly blending the Eulerian nature of NeRF with MPM's Lagrangian particle dynamics. (Section 3.3).

- Our framework estimates *both* the geometric structure and physical parameters of a wide variety of complex systems, including elastic materials, plasticine, sand, and Newtonian/non-Newtonian fluids, outperforming state-of-the-art approaches by up to **two orders of magnitude**. (Section 5).

## 2 RELATED WORK

**Neural radiance fields (NeRF)**, introduced in Mildenhall et al. (2020), are a widely adopted technique to encode scene geometry in a compact neural network; enabling photo-realistic rendering and depth estimation from novel views. A comprehensive survey of neural fields is available in Xie et al. (2022). In this work, we adopt the voxel representation proposed by Sun et al. (2022) as these do not require positional information and naturally fit the Eulerian stage of the Material Point Method (MPM) used in our physics prior.

For **perception of dynamic scenes**, Li et al. (2021) introduce forward and backward motion fields to enforce consistency in the representation space of neighboring frames. D-NeRF (Pumarola et al., 2021) introduces a canonical frame with a unique neural field for densities and colors, and a time-dependent backward deformation map to query the canonical frame. This representation has since been adopted in Tretschk et al. (2021) and Park et al. (2021). Chu et al. (2022) targets smoke scenes and advects the density field by the velocity field of smoke. This method does not deal with boundary conditions, so it cannot model solids and contact. Guan et al. (2022) present a combination of NeRF with intuitive fluid dynamics leveraging neural simulators; whereas we provided a principled, and interpretible simulation-and-rendering framework.

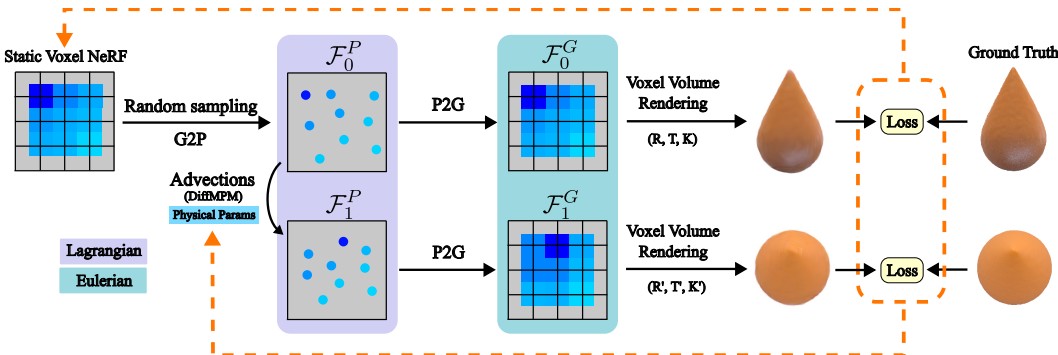

Figure 1: **PAC-NeRF** uses both Lagrangian (particle; material-space) and Eulerian (grid; world-space) representations for an accurate yet tractable computational model of continuum materials. P2G and G2P denote particle-to-grid and grid-to-particle transforms, respectively. Renderable quantities (volume densities, colors) are represented in the world space (first frame) using a voxel NeRF (Sun et al., 2022). These quantities are bound to particles (by a sampling scheme) whose dynamics are simulated by using a differentiable material point method (MPM) (Jiang et al., 2015). The (Eulerian) voxel representation enables efficient collision handling and rendering. Since the entire simulation and rendering pipeline are differentiable, rendered (color) images are able to optimize both the geometric and physical properties of objects. PAC-NeRF (1) accelerates NeRF rendering with Eulerian representation, (2) lends physical plausibility, interpretability, and data efficiency in dynamic scenes, and (3) enables physical parameter estimation for continuum materials.

**System identification of soft bodies** is an extremely challenging task, owing to its high dimensionality and the presence of large deformations. Neural (Sanchez-Gonzalez et al., 2020; Li et al., 2019; Xu et al., 2019) or gradient-free methods (Wang et al., 2015; Takahashi & Lin, 2019) struggle to achieve high accuracy on these problems, owing to their black-box nature. Recent progress in differentiable physics simulation has demonstrated great promise Qiao et al. (2021); Du et al. (2021); Rojas et al. (2021); Geilinger et al. (2020); Heiden et al. (2021); Jatavallabhula et al. (2020); Ma et al. (2021), but assumes that watertight mesh of objects are available. Recently, Chen et al. (2022); Qiao et al. (2022) estimate elastic object properties from videos. Our hybrid representation and simulation-rendering is general, and speeds up neural rendering by *two orders of magnitude*.

We use the **Material Point Method (MPM)** due to its ability to handle topology changes and frictional contacts; allowing the simulation of a wide range of materials, including elastic objects, sand (Klár et al., 2016), to fluids (Jiang et al., 2015) and foam (Yue et al., 2015). While MPM has previously been used for differentiable physics simulation (Hu et al., 2020; Huang et al., 2021; Fang et al., 2022), a watertight, non-degenerate mesh model was assumed to be available. Our method solves the long-standing challenge of perceiving geometric and physical properties solely from videos.

## 3 METHOD

**Problem specification**: Given a set of (posed) multi-view videos of a dynamic scene, we aim to recover (1) an explicit geometric representation, and (2) physical properties (such as Young's modulus, fluid viscosity, friction angles, etc.) of the dynamic object of interest.

Unlike existing system identification methods that operate on images, we do not require known object geometries. Our approach is general and works on a wide range of material types (fluids, sand, plasticine, etc.). Our proposed approach, Physics Augmented Continuum Neural Radiance Fields **PAC-NeRF**, seamlessly blends neural scene representations and explicit differentiable physics engines for continuum materials. The core components of PAC-NeRF include a continuum NeRF, a particle-grid interconverter, and a Lagrangian field. We detail these in this section (see Figure 1).

### 3.1 CONTINUUM NEURAL RADIANCE FIELDS

Recall that a (static) NeRF comprises a view-independent volume density field $\sigma(\mathbf{x})$ and a view-dependent appearance (color) field $\mathbf{c}(\mathbf{x}, \omega)$ for each point $\mathbf{x} \in \mathbb{R}^3$, and directions $\omega = (\theta, \phi) \in \mathbb{S}^2$ (spherical coordinates). A *dynamic* (time-dependent) NeRF extends the fields above with an additional time variable $t \in \mathbb{R}^+$, denoted $\sigma(\mathbf{x}, t)$ and $\mathbf{c}(\mathbf{x}, \omega, t)$ respectively. We use the efficient

voxel discretization from Sun et al. (2022) to specify a dynamic Eulerian (world-frame) NeRF. Color images are rendered from the above time-dependent fields by sampling points along a ray, for each pixel in the resultant image. The appearance $\mathbf{C}(\mathbf{r}, t)$ of a pixel specified by ray direction $\mathbf{r}(s)$ ($s \in [s_{\min}, s_{\max}]$) is given by the volume rendering integral (Mildenhall et al., 2020)

$$\mathbf{C}(\mathbf{r}, t) = \int_{s_n}^{s_f} T(s, t)\, \sigma(\mathbf{r}(s), t)\, \mathbf{c}(\mathbf{r}(s), \omega, t)\, ds + \mathbf{c}_{bg}\, T(s_f, t), \quad T(s, t) = \exp\left(-\int_{s_n}^{s} \sigma(\mathbf{r}(\bar{s}), t)\, d\bar{s}\right)$$
(1)

The dynamic NeRF can be trained by enforcing the rendered pixel colors to match those in the video.

$$\mathcal{L}_{render} = \frac{1}{N} \sum_{i=0}^{N-1} \frac{1}{|\mathcal{R}|} \sum_{\mathbf{r} \in \mathcal{R}} \|\mathbf{C}(\mathbf{r}, t_i) - \hat{\mathbf{C}}(\mathbf{r}, t_i)\|^2,$$
(2)

where $N$ is the number of frames of videos, $\hat{\mathbf{C}}(\mathbf{r}, t)$ is the ground truth color observation.

Additionally, we enforce that the appearance and volume density fields admit conservation laws characterized by the velocity field of the underlying **physical** system:

$$\frac{D\sigma}{Dt} = 0, \quad \frac{D\mathbf{c}}{Dt} = 0,$$
(3)

with $\frac{D\phi}{Dt} = \frac{\partial \phi}{\partial t} + \mathbf{v} \cdot \nabla \phi$ being the material derivative of an arbitrary time-dependent field $\phi(\mathbf{x}, t)$. Here, $\mathbf{v}$ is a *velocity field*, which in turn must obey momentum conservation for continuum materials

$$\rho \frac{D\mathbf{v}}{Dt} = \nabla \cdot \boldsymbol{T} + \rho \mathbf{g},$$
(4)

where $\rho$ is the physical density field, $\boldsymbol{T}$ is the internal Cauchy stress tensor, and $\mathbf{g}$ is the acceleration due to gravity. We use the differentiable Material Point Method (Hu et al., 2020) to evolve Equation (4).

## 3.2 PARTICLE-GRID INTERCONVERTERS

While a Lagrangian representation is ideal for advection by the material point method (MPM), an Eulerian frame is required for rendering the advected particle states to image space. We therefore employ a hybrid representation to blend the best of both worlds. A key requirement is to be able to seamlessly traverse the Eulerian (grid) view to the Lagrangian (particle) view (and vice versa).

Denoting the Eulerian and Lagrangian views $G$ and $P$ respectively, a field $\mathcal{F}_*^G(t) = \{\sigma(\mathbf{x}, t), (\mathbf{x}, t)\}$ at time $t$ may be interconverted as follows:

$$\mathcal{F}_p^P \approx \sum_i w_{ip} \mathcal{F}_i^G, \quad \mathcal{F}_i^G \approx \frac{\sum_p w_{ip} \mathcal{F}_p^P}{\sum_p w_{ip}},$$
(5)

where $i$ indices grid nodes and $p$ indices particles, and $w_{ip}$ is the weight of the trilinear shape function defined on node $i$ and evaluated at the location of particle $p$. We use *P2G* and *G2P* to denote the particle-to-grid and grid-to-particle conversion processes respectively.

## 3.3 LAGRANGIAN FIELD

An Eulerian voxel field $\mathcal{F}^G(t_0)$ is initialized over the first frame of the sequence. From this field, we use the *G2P* process to obtain a Lagrangian particle field $\mathcal{F}^P(t_0)$. We advect this field using an initial guess physical parameter set $\boldsymbol{\Theta}$ and the material point method (Equation (3)) to obtain $\mathcal{F}^P(t_1)$ at $t_1 = t_0 + \delta t$ (where $\delta t$ is the duration of each simulation timestep). The advected field is then mapped to the Eulerian view using the *P2G* process, resulting in $\mathcal{F}^G(t_1)$, which is employed for collision handling and neural rendering. The Eulerian voxel grid representation used here is at least two orders of magnitude ($100\times$) faster in terms of image rendering time (Sun et al., 2022).

Following Sun et al. (2022), the rendering density field $\sigma$ and color field $\mathbf{c}$ at time $t$ are:

$$\sigma(\mathbf{x}, t) = \text{softplus}(\text{Interp}(\mathbf{x}, \hat{\sigma})), \quad \mathbf{c}(\mathbf{x}, \mathbf{d}, t) = \text{MLP}(\text{Interp}(\mathbf{x}, \hat{\mathbf{c}}), \mathbf{d}),$$
(6)

where $\hat{\sigma}$ is a scalar field and $\hat{\mathbf{c}}$ is a vector field, both are discretized on a fixed voxel grid. $\text{Interp}(\cdot)$ denotes trilinear interpolation. Advection is performed by first advecting the grids $\hat{\sigma}$ and $\hat{\mathbf{c}}$, followed by computing the interpolation functions and evaluating the MLP (or the softplus).

The initialization of forward simulation requires generating Lagrangian representations of $\hat{\sigma}$ and $\hat{\mathbf{c}}$. We randomly sample 8 particles within each voxel grid and use Equation (5) to bind the density and color values onto particles. Additionally, we associate with each particle a scalar value $\alpha_p = 1 - e^{-\text{softplus}(\hat{\sigma}_p)}$ in $(0, 1)$. A lower $\alpha$ value denotes a smaller contribution to the radiance fields. We make particles with lower values of $\alpha$ softer by scaling the physical density field $\rho$ (Equation (4)) and the physical stress field $\boldsymbol{T}$ by a factor $\alpha^3$. We remove a particle $p$ if $\alpha_p < \epsilon \max_p \alpha_p$, where we set $\epsilon = 10^{-3}$ as a constant threshold.

## 3.4 PAC-NeRF FOR GEOMETRY-AGNOSTIC SYSTEM IDENTIFICATION

Our pipeline for geometry-agnostic system identification comprises three distinct phases for computational tractability. We first preprocess data using video matting techniques to extract foreground objects of interest. This is followed by a geometry seeding phase, where we employ a coarse-to-fine approach to recover object geometry. The extracted geometry is then used to perform system identification by rendering out future video frames based on a guessed set of physical properties, computing an error term with respect to the true videos, and updating the physical properties by gradient-based optimization.

**Data Preprocessing**: We assume a set of static cameras with known intrinsics and extrinsics. To focus rendering computation on the object of interest, we run the video matting framework in Lin et al. (2021) to remove static background objects. This also provides us with a segmentation mask of the foreground object of interest.

**Geometry Seeding**: We first obtain a coarse geometry of the foreground object(s) of interest by employing the static voxel fields. The contents of the foreground segmentation mask are rendered to produce an appearance loss term optimized using gradient descent. We employ a coarse-to-fine strategy to enable easier optimization. We noticed that, as opposed to directly rendering the initial voxel radiance field, running the G2P converter followed by a P2G converter ensures that rendering for all the frames (including the initial frame) is consistently based on the same group of particles (implicitly providing a set of consistent correspondences across time). In addition to the rendering loss Equation (2), we employ a surface regularizer to regularize the geometric density field:

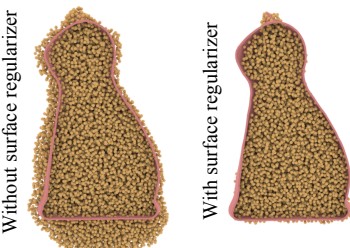

Figure 2: Our surface regularizer improves reconstruction quality, by producing a tight-fit shape to the segmentation mask.

$$\mathcal{L}_{\text{surf}} = \sum_p \text{clamp}(\alpha_p, 10^{-4}, 10^{-1})(\frac{\Delta x}{2})^2. \tag{7}$$

This regularizer minimizes the total surface area, and as shown in Figure 2, this tends to improve the quality of reconstructed geometries by making the reconstructed point cloud more compact and fit the ground truth boundary more closely.

**System Identification**: Upon convergence of the geometry seeding phase, we freeze the parameters of the field (for the initial time step $t_0$). To minimize scenarios where one or more variables of interest become unobservable under unknown initial conditions, we use the first 2-3 frames to estimate the initial velocity of observed particles. We then optimize for physical parameters with each subsequent frame. To mitigate convergence issues in parameter spaces with larger degrees of freedom, we found it helpful to warm start the optimizer after initial collision events, followed by an optimization over the entire sequence.

## 4 IMPLEMENTATION DETAILS

The architecture of voxel discretization of NeRF follows Sun et al. (2022), which stores density value and color feature within $160^3$ voxels, only contraining an extra 2-layer MLP with a hidden dimension 128 for view-dependent color fields. The dimension of color feature on each voxel is 12. Positional embedding is applied to the inputs (query position, view direction and color feature) to the shallow MLP, leading to a input dimension 39. Our differentiable MPM is using DiffTaichi (Hu et al., 2020). Both the rendering and simulation are optimized for GPU. The entire training

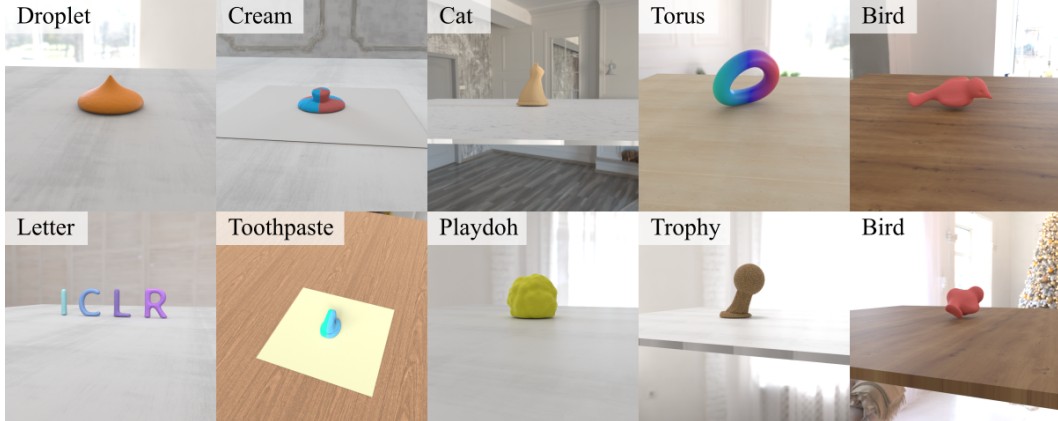

Figure 3: The **photorealistic dataset** used to evaluate system identification and geometry estimation. The dataset includes a variety of continuum materials, including Newtonian fluids (Droplet, Letter), non-Newtonian fluids (Cream, Toothpaste), granular media (Trophy), deformable solids (Torus, Bird), and plasticine (Cat, Playdoh). All objects freely fall under the influence of gravity, undergoing collisions. Objects are rendered under complex environmental lighting conditions for photorealism.

takes $\sim 1.5$ hours on a single Nvidia 3090 GPU. Simulation+rendering of one frame takes $\sim 1s$ (vs. $\sim 10min$ for Chen et al. (2022)).

## 5 EXPERIMENTS

We conduct various experiments to study the efficacy of PAC-NeRF on system identification tasks and find that:

- PAC-NeRF can recover high-quality object geometries solely from videos.
- PAC-NeRF performs significantly better on system identification tasks compared to fully learned approaches.
- PAC-NeRF alleviates the assumptions that other techniques require (i.e., known object geometry), while outperforming them.
- Purely pixel-based loss functions provide rich gradients that enable physical parameter estimation.

### 5.1 EXPERIMENT SETUP

**Dataset**: To evaluate different system identification methods, we simulate and render a wide range of objects using a photo-realstic simulation engine with varying environment lighting conditions and ground textures. Our dataset includes deformable objects, plastics, granular media, Newtonian and non-Newtonian fluids. Figure 3 demonstrates a few example scenarios. In each scene, objects freely fall under the influence of gravity, undergoing (self and external) collisions. For convenience, we assume that the collision objects such as the ground plane are known (however, this can also be easily estimated from observed images). Each scene is captured from 11 uniformly sampled viewpoints, with the cameras evenly spaced on the upper hemisphere containing the object. All ground truth simulation data are generated by MLS-MPM framework (Hu et al., 2018b).

**Physical Parameters**: Our differentiable MPM implementation supports five kinds of common materials, including elasticity, plasticine, granular media (e.g., sand), Newtonian fluids and non-Newtonian fluids.

- *Elasticity*: Young's modulus ($E$) (material stiffness), Poisson's ration ($\nu$) (ability to preserve volume under deformation).
- *Plasticine*: Young's modulus ($E$), Poisson's ration ($\nu$), and yield stress ($\tau_Y$) (stress required to cause permanent deformation/yielding).
- *Newtonian fluid*: fluid viscosity ($\mu$) (opposition to velocity change), bulk modulus ($\kappa$) (ability to preserve volume).

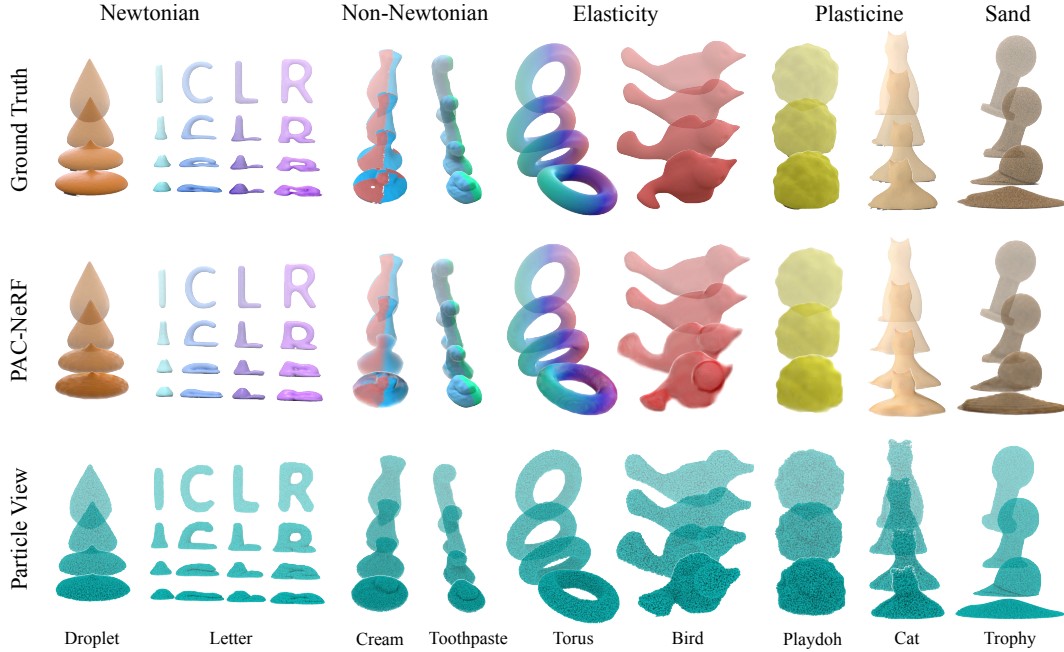

Figure 4: **Qualitative results**: Our experiments cover a wide range of continuums, from Newtonian fluids and non-Newtonian fluids, to elastic/plastic solids and granular media (sand). Existing approaches (including dynamic NeRF and variants) are unable to reconstruct these highly dynamic objects. PAC-NeRF not only reconstructs them with high accuracy but also precisely determines the underlying physical properties.

| | Initial Guess | Optimized | Ground Truth |
|---|---|---|---|
| Droplet | $\mu = 1, \kappa = 10^3$ | $\mu = 2.09 \times 10^2, \kappa = 1.08 \times 10^5$ | $\mu = 200, \kappa = 10^5$ |
| Letter | $\mu = 1, \kappa = 10^6$ | $\mu = 83.85, \kappa = 1.35 \times 10^5$ | $\mu = 100, \kappa = 10^5$ |
| Cream | $\mu = 10^2, \kappa = 10^4, \tau_Y = 10, \eta = 1$ | $\mu = 1.21 \times 10^5, \kappa = 1.57 \times 10^6, \tau_Y = 3.16 \times 10^3, \eta = 5.6$ | $\mu = 10^4, \kappa = 10^6, \tau_Y = 3 \times 10^3, \eta = 10$ |
| Toothpaste | $\mu = 10^2, \kappa = 10^4, \tau_Y = 10, \eta = 1$ | $\mu = 6.51 \times 10^3, \kappa = 2.22 \times 10^5, \tau_Y = 228, \eta = 9.77$ | $\mu = 5 \times 10^3, \kappa = 10^5, \tau_Y = 200, \eta = 10$ |
| Torus | $E = 10^5, \nu = 0.1$ | $E = 1.04 \times 10^6, \nu = 0.322$ | $E = 10^6, \nu = 0.3$ |
| Bird | $E = 10^3, \nu = 0.1$ | $E = 2.78 \times 10^5, \nu = 0.273$ | $E = 3 \times 10^5, \nu = 0.3$ |
| Playdoh | $E = 10^5, \nu = 0.4, \tau_Y = 10^3$ | $E = 3.84 \times 10^6, \nu = 0.272, \tau_Y = 1.69 \times 10^4$ | $E = 2 \times 10^6, \nu = 0.3, \tau_Y = 1.54 \times 10^4$ |
| Cat | $E = 10^3, \nu = 0.2, \tau_Y = 10^2$ | $E = 1.61 \times 10^5, \nu = 0.293, \tau_Y = 3.57 \times 10^3$ | $E = 10^6, \nu = 0.3, \tau_Y = 3.85 \times 10^3$ |
| Trophy | $\theta_{fric}^0 = 10°$ | $\theta_{fric}^0 = 36.1°$ | $\theta_{fric}^0 = 40°$ |

Table 1: PAC-NeRF estimates physical parameters very accurately.

- *Non-Newtonian fluid*: shear modulus ($\mu$), bulk modulus ($\kappa$), yield stress ($\tau_Y$), and plasticity viscosity ($\eta$) (decayed temporary resistance to yielding).
- *Sand*: friction angle ($\theta_{fric}$) (proportionality constant determining slope of a sand pile).

**System Identification Pipeline**: We first train a static voxel NeRF using data from the first frame (following Sun et al. (2022)) with the Adam optimizer. The initial velocity estimator uses L-BFGS, which we experimentally find to be better than Adam for this sub-task. For all other physical parameters of interest, we use the Adam optimizer.

## 5.2 SYSTEM IDENTIFICATION EVALUATION

**Synthetic Data**: We report system identification results using PAC-NeRF over 9 synthetic problem instances. The qualitative results are shown in Figure 4. And the quantitative results are listed in Table 1. Each row lists the initial guess, the values obtained after optimization, and the ground truth. We see that, in each case, the optimized physical parameters closely agree with the ground truth.

**Real Data**: We also evaluate PAC-NeRF on real-world data (Figure 5). We built a capture system comprising four synchronized Intel RealSense D455 cameras capable of streaming RGB images at a resolution of $640 \times 480$ and at a rate of 60 frames per second. Note that we DO NOT use any depth data recorded from these cameras in our experiments, for fair evaluation. We captured a deformable ball falling onto a table and manually segmented the data. Our reconstructed scene qualitatively matches the target videos. We note that, due to the limited number of views (here, 4) we could

Figure 5: We test PAC-NeRF on a set of real-world multiview videos. Our reconstructed scene qualitatively matches the target videos, which validates that the rendering loss is effectively minimized.

acquire, there are slight errors in surface geometry estimation. These issues may be mitigated by increasing the number of views from which data is captured.

## 5.3 COMPARISONS

Since there are no existing approaches that estimate *both* geometry and material properties from videos, we make a best-effort comparison by blending combinations of state-of-the-art approaches for each sub-task. For each material type discussed above, we generate 10 problem instances by varying object orientations, initial velocities, and physical parameters. The comparisons are conducted on this dataset. Note that our approach (and most of the baselines herein) does not require a separate training phase – they perform inference-time optimization on each test sequence.

### 5.3.1 APPROACHES EVALUATED

**Multi-view LSTM**: To evaluate amortized physical parameter inference schemes, we use a pre-trained ResNet feature extractor to extract features from all views and feed them into a 2-layer LSTM. This baseline uses privileged information in the form of training video sequences (while all other baselines begin with a random initialization on each test sequence).

**D-NeRF + DiffSim**: To assess the impact of conservation law enforcement in PAC-NeRF, we compare with VEO (Chen et al., 2022) (a best-effort comparison) by implementing D-NeRF (dynamic NeRF) and integrating it with our differentiable simulator. A dynamic voxel NeRF learns both the forward and backward deformation field for each frame. The learned forward deformation field is then used to provide the differentiable simulator with 3D supervision.

**NeRF + ∇Sim**: We also compare with ∇Sim – a state-of-the-art approach for estimating physical properties with both the geometry and rendering configuration known. (By contrast, we assume neither.) We implement a best-effort comparison where we use a static voxel NeRF to recover the geometry and directly provide the rendering color configuration. Note that ∇Sim *only supports FEM simulation* for continuum materials. Therefore, this baseline only runs on a small subset of our scenarios. Moreover, to run this baseline, we extract a surface mesh from our reconstructed point cloud from voxel NeRF and then use TetWild (Hu et al., 2018a) to generate a tetrahedral mesh. Further, we manually choose an optimal camera viewing direction to facilitate optimization.

**Random**: We also evaluate the performance of these approaches against random chance, by employing a baseline that predicts a uniformly randomly chosen value over the parameter ranges used.

**Oracle**: We implement an oracle that uses privileged 3D geometric information. The oracle knows the ground truth point clouds, and employs 3D supervision (Chamfer distance) to infer physical properties. Of the 10 problem instances for each material type, we run the oracle on 4.

### 5.3.2 ANALYSIS OF RESULTS

The performances of all approaches on the system identification tasks are listed in Table 2. For each material, we report the mean absolute error and its standard deviation over the evaluated instances. We see that compared to all other methods, PAC-NeRF achieves the best results in 14 of the 17 categories of physical properties estimated.

The multi-view LSTM method is adapted from DensePhysNet (Xu et al., 2019) and learns the mapping from videos to physical parameters implicitly. Notably, this baseline requires privileged information in the form of training sequences – not available to the other approaches.

The D-NeRF+Diffsim is adapted from VEO (Chen et al., 2022), where the forward deformation field is used to optimize a differentiable simulation. The accuracy of this approach therefore relies on the quality of the learned forward deformation field. However, this learned deformation field (Tretschk

| | | Ours | Multi-view LSTM | D-NeRF + DiffSim | NeRF + $\nabla$Sim | Random | Oracle ($\times 10^{-2}$) |
|---|---|---|---|---|---|---|---|
| Newtonian | $\log_{10}(\mu)$ | **11.6 ± 6.60** | 14.1 ± 9.26 | > 291.3 | | 64.1 ± 31.3 | 19.4 ± 6.95 |
| | $\log_{10}(\kappa)$ | **16.7 ± 5.37** | 28.2 ± 19.8 | > 85.2 | Not supported | 72.8 ± 42.5 | 258.9 ± 25.2 |
| | $v$ | **0.86 ± 1.45** | 23.1 ± 17.8 | 25.5 | | 55.5 ± 22.6 | 2.82 ± 1.72 |
| Non-Newtonian | $\log_{10}(\mu)$ | **24.1 ± 21.9** | 39.4 ± 26.9 | > 276.7 | | 34.2 ± 19.9 | **24.0 ± 17.1** |
| | $\log_{10}(\kappa)$ | 44.0 ± 26.3 | **25.1 ± 22.6** | > 262.5 | | 67.6 ± 49.4 | 82.90 ± 21.0 |
| | $\log_{10}(\tau_Y)$ | **5.09 ± 7.41** | 7.19 ± 7.88 | > 359.6 | Not supported | 30.0 ± 15.0 | 9.75 ± 11.3 |
| | $\log_{10}(\eta)$ | **28.7 ± 23.3** | 39.9 ± 21.1 | > 86.1 | | 64.3 ± 43.7 | 90.6 ± 37.2 |
| | $v$ | **0.29 ± 0.13** | 24.4 ± 14.2 | 25.9 | | 52.2 ± 21.6 | 2.60 ± 0.39 |
| Elasticity | $\log_{10}(E)$ | **3.02 ± 3.72** | 17.7 ± 9.25 | > 437.5 | 151.6 ± 42.6 | 96.2 ± 46.9 | 4.39 ± 4.54 |
| | $\nu$ | **4.35 ± 5.08** | 81.8 ± 58.4 | 7.67 | 16.4 ± 6.58 | 10.9 ± 6.37 | 3.65 ± 2.88 |
| | $v$ | **0.50 ± 0.23** | 6.10 ± 3.27 | 30.7 | 182.4 ± 74.1 | 43.6 ± 25.4 | 2.69 ± 0.97 |
| Plasticine | $\log_{10}(E)$ | 83.8 ± 68.4 | 41.1 ± 31.4 | **23.2** | | 42.9 ± 38.0 | 56.2 ± 30.7 |
| | $\log_{10}(\tau_Y)$ | **11.2 ± 14.5** | 28.6 ± 17.3 | > 268.9 | Not supported | 82.7 ± 43.1 | 8.13 ± 3.61 |
| | $\nu$ | 18.9 ± 15.7 | **5.40 ± 3.49** | 10.36 | | 10.6 ± 6.15 | **4.44 ± 3.61** |
| | $v$ | **0.56 ± 0.17** | 22.0 ± 14.8 | > 384.6 | | 47.7 ± 17.53 | 4.06 ± 1.87 |
| Sand | $\theta_{fric}$ | **4.89 ± 1.10** | 20.1 ± 2.06 | 64.5 | Not supported | 22.5 ± 14.7 | **0.44 ± 0.42** |
| | $v$ | **0.21 ± 0.08** | 14.6 ± 6.25 | 40.6 | | 38.7 ± 21.1 | 1.17 ± 0.42 |

Table 2: **System identification performance**: Means and the standard derivations of absolute errors are reported for each metric. We compare with five baselines, including a pure vision-based method (ResNet+LSTM regression), D-NeRF+DIffSim (similar to VEO (Chen et al., 2022)), NeRF+$\nabla$Sim (Jatavallabhula et al., 2020), random sampling from parameter distribution, and oracle point cloud plus Chamfer distance minimization. Our method obtains the best results (highlighted in boldface font) in 14/17 entries, excluding the Oracle.

et al., 2021) cannot guarantee physical correctness, unlike PAC-NeRF which is constrained by the conservation laws (Equation (3) and Equation (4)). In our experiments, we observed that this leads to very noisy deformation, and impedes system identification performance. Due to these instabilities of this approach, we only run this on one scenario per material type and report its performance.

The NeRF+$\nabla$Sim (Jatavallabhula et al., 2020) baseline only supports FEM simulations for elastic materials and is sensitive to time integration step size. To stabilize the symplectic FEM and the contact model used in $\nabla$Sim, tens of thousands of substeps are required to simulate the entire sequence. Notwithstanding the errors induced due to geometric inaccuracies and unknown rendering configurations, errors accumulated from long timestepping sequences also contribute to the failure of $\nabla$Sim on our scenarios. PAC-NeRF, with its Eulerian-Lagrangian representation, is more robust under larger deformations (e.g., fluids and sand) and permits larger time step sizes than a FEM simulation with a symplectic integrator (used in $\nabla$Sim).

The Oracle baseline assumes known 3D (point cloud) geometry and computes a Chamfer distance error term per-timestep. We note, surprisingly, that in several scenarios, a pixel-wise color loss outperforms this 3D supervision signal. The results show the effectiveness of the differentiable rendering-simulation pipeline, where a 2D pixel-level supervision better optimizes the physical parameters, opposed to a 3D Chamfer distance metric.

## 6 CONCLUSION

We presented Physics Augmented Continuum Neural Radiance Fields (PAC-NeRF), a novel representation for system identification without geometry priors. The continuum reformulation of dynamic NeRF naturally unifies physical simulation and rendering, enabling differentiable simulators to estimate both geometry and physical properties from image space signals (such as multi-view videos). A hybrid representation is used to discretize our PAC-NeRF implementation, which leverages the high efficiency of an Eulerian voxel-based NeRF and the flexibility of MPM.

Despite the promising results, this work has some limitations. It assumes the availability of synchronized and accurately calibrated cameras to ensure good-quality reconstruction using NeRF. The scenes used in this work should also be easily amenable to video matting (background removal). Moreover, this work assumes the underlying physical phenomenon follows continuum mechanics and cannot automatically distinguish between different materials. Future work could focus on extending the MPM framework beyond volumetric continuum materials, such as cloth, using neural constitutive models (Li et al., 2022), implicit MPM for stiff materials, and integrating other differentiable simulators like articulated body simulators. Additionally, incorporating interactions with rigid bodies could enable manipulation tasks (Huang et al., 2021) on NeRF-represented soft-bodies.

ACKNOWLEDGMENTS

This project was supported by the DARPA MCS program, MIT-IBM Watson AI Lab, NSF CAREER 2153851, CCF2153863, ECCS-2023780, and gift funding from MERL, Cisco, and Amazon.

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

## A APPENDIX

### A.1 PHYSICAL MODELS

MPM supports all kinds of constitutive models and plasticity models. In continuum physics framework, the internal force is provided by Cauchy stress $\mathbf{T}$, a tensor field defined by deformation gradient $\mathbf{F}$. The deformation gradient is tracked on MPM particles to measure their distortion compared to its initial state. For plasticity, the deformation gradient is constrained within an elastic region. A return mapping $\mathcal{Z}$ is applied to pull deformation gradient back onto the yield surface.

**Elasticity** We use neo-Hookean elasticity to model elastic objects. The Cauchy stress for this model is defined by

$$J\mathbf{T}(\mathbf{F}) = \mu(\mathbf{F}\mathbf{F}^\top) + (\lambda \log(J) - \mu)\mathbf{I}, \tag{8}$$

where $J = \det(\mathbf{F})$. $\mu, \lambda$ are Lame parameters, related to Young's modulus ($E$) and Poisson's ratio ($\nu$) by

$$\mu = \frac{E}{2(1+\lambda)}, \quad \lambda = \frac{\nu E}{(1+\nu)(1-2\nu)}. \tag{9}$$

**Newtonian Fluid** We use J-based fluid combined with viscosity term to model Newtonian fluid. The stress for this model is defined by

$$J\mathbf{T}(\mathbf{F}) = \frac{1}{2}\mu(\nabla\mathbf{v} + \nabla\mathbf{v}^\top) + \kappa(J - \frac{1}{J^6}). \tag{10}$$

**Plasticine** To simulate plasticine, we use St.Venant-Kirchhoff (StVK) constitutive model combine with von-Mises plastic return mapping. The stress associated with the model is defined by

$$J\mathbf{T}(\mathbf{F}) = \mathbf{U}(2\mu\boldsymbol{\epsilon} + \lambda \operatorname{Tr}(\boldsymbol{\epsilon}))\mathbf{U}^\top, \tag{11}$$

with $\mathbf{F} = \mathbf{U}\boldsymbol{\Sigma}\mathbf{V}$ being the SVD decomposition of $\mathbf{F}$ and $\boldsymbol{\epsilon} = \log(\boldsymbol{\Sigma})$ being the Hencky strain.

The von-Mises yielding condition is

$$\delta\gamma = \|\hat{\boldsymbol{\epsilon}}\| - \frac{\tau_Y}{2\mu} > 0, \tag{12}$$

where $\hat{\epsilon}$ is the normalized Hencky strain. $\tau_Y$ is the yield stress, which is the parameter for this model. $\delta\gamma > 0$ means the deformation gradient is outside the elastic region. The deformation gradient will be projected back onto the boundary of elastic region (yield surface) by the following return mapping:

$$\mathcal{Z}(\mathbf{F}) = \begin{cases} \mathbf{F} & \delta\gamma \leq 0 \\ \mathbf{U}\exp\left(\boldsymbol{\epsilon} - \delta\gamma\frac{\hat{\boldsymbol{\epsilon}}}{\|\hat{\boldsymbol{\epsilon}}\|}\right)\mathbf{V}^\top & \text{otherwise} \end{cases}. \tag{13}$$

**Non-Newtonian Fluid** Non-Newtonian fluid also has a yield stress. We use viscoplastic model (Yue et al., 2015) to model non-Newtonian fluid. We still use von-Mises criteria to define elastic region. However, the existence of viscoplastic means the deformation will not be directly projected back onto the yield surface. Define

$$\hat{\mu} = \frac{\mu}{d}\text{Tr}(\boldsymbol{\Sigma}^2)$$
$$\boldsymbol{s} = 2\mu\hat{\epsilon} \tag{14}$$
$$\hat{s} = \|\boldsymbol{s}\| - \frac{\delta\gamma}{1 + \frac{\eta}{2\hat{\mu}\Delta t}}$$

$$\mathcal{Z}(\mathbf{F}) = \begin{cases} \mathbf{F} & \delta\gamma \leq 0 \\ \mathbf{U}\exp\left(\frac{\hat{s}}{2\mu}\hat{\epsilon} + \frac{1}{d}\text{Tr}(\boldsymbol{\epsilon})\mathbf{1}\right)\mathbf{V}^\top & \text{otherwise} \end{cases}. \tag{15}$$

**Sand** Drucker-Prager yield criteria is used to simulate granular materials (Klár et al., 2016). The underlying constitutive model is still StVK. The yielding criteria is defined by

$$\text{tr}(\boldsymbol{\epsilon}) > 0, \quad \text{or} \quad \delta\gamma = \|\hat{\epsilon}\|_F + \alpha\frac{(d\lambda + 2\mu)\text{tr}(\boldsymbol{\epsilon})}{2\mu} > 0. \tag{16}$$

where $\alpha = \sqrt{\frac{2}{3}}\frac{2\sin\theta_{fric}}{3-\sin\theta_{fric}}$ and $\theta_{fric}$ is the friction angle. We use the following return mapping in sand simulations:

$$\mathcal{Z}(\mathbf{F}) = \begin{cases} \mathbf{U}\mathbf{V}^\top & \text{tr}(\boldsymbol{\epsilon}) > 0 \\ \mathbf{F} & \delta\gamma \leq 0, \text{tr}(\boldsymbol{\epsilon}) \leq 0 \\ \mathbf{U}\exp\left(\boldsymbol{\epsilon} - \delta\gamma\frac{\hat{\boldsymbol{\epsilon}}}{\|\hat{\boldsymbol{\epsilon}}\|}\right)\mathbf{V}^\top & \text{otherwise} \end{cases}. \tag{17}$$

## B COMPLEX BOUNDARY CONDITIONS

Pre-known boundary conditions are trivially to be added in MPM simulations, just like the ground. Here we shown examples of elastic rope falling onto two rigid cylinders in Figure 6. The initial guess, optimized values and the ground truth values of physical parameters are listed in Table 3.

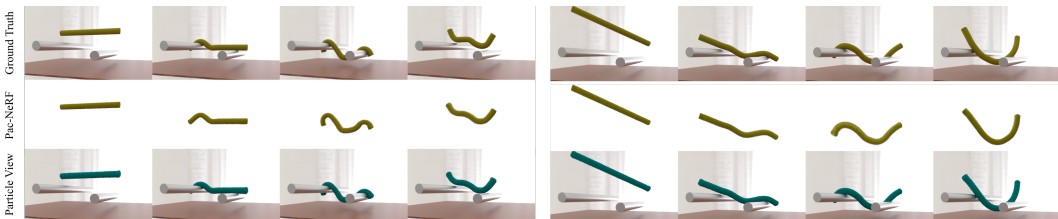

Figure 6: Examples of elastic ropes falling onto cylinders. The top row is the ground truth video. PAC-NeRF is trained on the segmented data, whose rendering outputs are visualized in the middle row. The bottom row is the reconstructed MPM particles.

## C RECONSTRUCTION QUALITY COMPARISON WITH D-NERF

When evaluating the D-NeRF+Diffsim baseline, we observe that D-NeRF is sensitive to sudden, large deformations (see Figure 7). As the simulation progresses, D-NeRF suffers from artificial

| | Initial Guess | Optimized | Ground Truth |
|---|---|---|---|
| Rope (Short) | $E = 10^3, \nu = 0.4$ | $E = 1.12 \times 10^5, \nu = 0.22$ | $E = 10^5, \nu = 0.3$ |
| Rope (Long) | $E = 10^4, \nu = 0.4$ | $E = 1.09 \times 10^6, \nu = 0.31$ | $E = 10^6, \nu = 0.3$ |

Table 3: Quantitative results of elastic rope example.

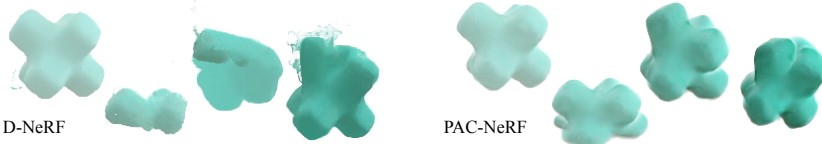

Figure 7: D-NeRF suffers from major artifacts as the object undergoes sudden large deformations. Notice how the object becomes partially absent in the middle two frames.

extreme deformation, resulting in the object breaking apart. This degrades the simulation to the extent that parts of the object even partially disappear (violating physical plausibility). This artifact is due to the learned backward deformation map points to an empty area in the canonical space, resulting in zero volume density. By contrast, our PAC-NeRF constrains forward deformations to follow the physical conservation law and does not experience this kind of unphysical result.

## D QUALITATIVE COMPARISON WITH ∇SIM ON REAL-WORLD DATA

We also tried ∇Sim on our real-world data. The qualitative comparison between PAC-NeRF and ∇Sim is shown in Figure 8. As we do in the baseline comparisons, we extract a surface mesh from our reconstructed point cloud from voxel NeRF and then generate a tetrahedral mesh, manually pick an optimal camera and set an approximate rendering configuration. As we discussed in our baseline comparison, explicit FEM used in ∇Sim suffers from tiny time steps, leading to significant numerical errors in backpropagations. Combined with errors from the geometry reconstructions and the approximate rendering configurations, all these factors contribute to the failure of ∇Sim.

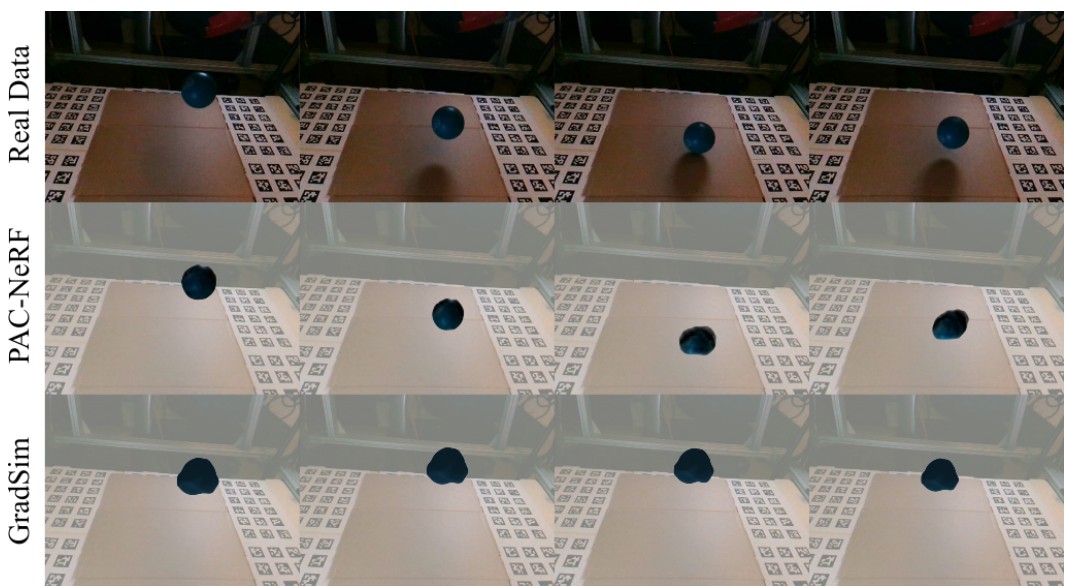

Figure 8: Qualitative comparison with ∇Sim on real-world data.

