# OpenReview forum: "PAC-NeRF: Physics Augmented Continuum Neural Radiance Fields for Geometry-Agnostic System Identification"
_ICLR.cc/2023/Conference — ICLR 2023 notable top 25%_

### Official Review · Reviewer_EVdM · 2022-10-19

**Confidence:** 4
**Correctness:** 3
**Technical Novelty And Significance:** 3
**Empirical Novelty And Significance:** 3
**Recommendation:** 6

**Clarity, Quality, Novelty And Reproducibility:**

This paper is well-written, and most results are promising. The idea is novel, and I think it is reproducible.

**Strength And Weaknesses:**

Strength:
(1) It is interesting to combine NeRF with physical constraints.
(2) This paper is well written.
(3) In most cases, the results are encouraging.

Summary of Weakness:
(1) Euler method and advection equation are both very fundamental and usually applied in highly restricted mathematical or physical models, which are not defined directly so that the theoretical level of this work satisfies a more general case.
(2) There are some limitations as mentioned before, which are not listed in the article clearly.
(3) Part of the experimental results is difficult to understand, disobeying universal physics phenomenon.
(4)The authors haven’t listed the source of their ground truth data.

Main Questions:
Though part of the results shown in this article reflects stable properties of the framework, the authors may not be familiar with the fluid dynamics field. Here are some following questions:
1) This article is aimed to estimate both the unknown geometry and physical parameters of highly dynamic objects from multi-view videos. It is noticed that a droplet is regarded as Newtonian fluid, but this property would be destroyed when velocity exceeds a boundary value. Additionally, in the general case, the descent of a droplet drop to the ground may not satisfy the two assumptions.
2) As for fluid dynamics fields, what is the source of the listed ground truth data (In house codes, CFD software, or open-source codes)? It cannot be made sure whether this ground truth data is convincing or not.
3) It is conspicuous that the method presented in this article requires the properties defined as continuum and conservation. Hence, listing the limitations clearly is necessary.

Things to improve the paper that did not impact the score:
(1) Clarify the limitations of the framework;
(2) If simulations of physics models are unavoidable, fundamental knowledge of other fields is necessary to present meaningful outcomes. For example, fluid dynamics is necessary for the authors to get familiar with.
(3) Illustrating what kinds of physics models could be solved by the Euler scheme and governed by the advection scheme with low accuracy loss and time steps cost.

**Summary Of The Paper:**

This article presents a novel system that brings differentiable physics and neural radiance fields for dynamic scenes by augmenting a NeRF with a differentiable continuum dynamics model. A hybrid Eulerian-Lagrangian formulation is defined in this system to permit advecting geometry and appearance. This framework outperforms previous approaches validated by a wide variety of systems.

**Summary Of The Review:**

I have listed my questions in Weakness. It is necessary to explain the results of droplets and the limitation of the proposed method. Overall, it is a nice conference paper.

---

> ### Author Response · Authors · 2022-11-18
> **Response to Reviewer EVdM**
>
> We thank the reviewer for the valuable feedback and positive comments. We address each of the concerns and questions below.
>
> > **Q1:** It is noticed that a droplet is regarded as Newtonian fluid, but this property would be destroyed when velocity exceeds a boundary value. Additionally, in the general case, the descent of a droplet drop to the ground may not satisfy the two assumptions.
>
> **A1:** We have listed it as a limitation that our framework cannot automatically change material types. Like existing system identification works [1, 2], the goal of our work is to find parameters for a known underlying parameterized physical system. The physical phenomena that our framework can interpret are restricted by the underlying simulator. For example, most honeys are Newtonian fluids; then we can use Newtonian fluid dynamics as the underlying physics to fit a **droplet** of honey falling onto the ground in general. If the inference results are poor, it means that the data is out of the scope of the presumed underlying physics.
>
> [1] Murthy, J. Krishna, et al. "gradSim: Differentiable simulation for system identification and visuomotor control." International Conference on Learning Representations. 2020.
>
> [2] Ma, Pingchuan, et al. “Risp: Rendering-invariant state predictor with differentiable simulation and rendering for cross-domain parameter estimation”. In International Conference on Learning Representations, 2021.
>
>
>
>
> > **Q2:** As for fluid dynamics fields, what is the source of the listed ground truth data (In house codes, CFD software, or open-source codes)? It cannot be made sure whether this ground truth data is convincing or not.
>
> **A2:** All of our ground truth simulations are generated by the MLS-MPM framework [3]. MPM is a universal framework for solid and fluid. It is a PDE-consistent method, which can converge to the true solution under refinement (continuously increasing simulation resolution) [4]. We have included the source of our ground truth data in the experiment setup section.
>
> [3] Hu, Yuanming, et al. "A moving least squares material point method with displacement discontinuity and two-way rigid body coupling." ACM Trans. Graph. 2018.
> [4] Sulsky, et al. “Application of a particle-in-cell method to solid mechanics”, Computer physics communications, 1995
>
> > **Q3:** It is conspicuous that the method presented in this article requires the properties defined as continuum and conservation. Hence, listing the limitations clearly is necessary.
>
> **A3:** We have listed this continuum assumption as a limitation in the revised manuscript. We will consider physical phenomena that cannot be easily modeled by continuum mechanics in future work, such as articulated bodies.
>
>
> > **Q4:** Clarify the limitations of the framework
>
> **A4:** Like NeRF, the performance of our framework highly depends on accurate camera parameters. As discussed above, we assume the underlying physical phenomenon follows continuum mechanics. We have included these limitations in our revision.
>
>
> > **Q5:** If simulations of physics models are unavoidable, fundamental knowledge of other fields is necessary to present meaningful outcomes. For example, fluid dynamics is necessary for the authors to get familiar with.
>
> **A5:** We followed the same fluid dynamics used in the MLS-MPM framework. Using more sophisticated physical models can further reduce the sim-to-real gap. We have since listed it as our future work.
>
>
> > **Q6:** Illustrating what kinds of physics models could be solved by the Euler scheme and governed by the advection scheme with low accuracy loss and time steps cost.
>
> **A6:** Almost all time-dependent physical procedure studied by continuum mechanics is described by the advection under the conservation of mass and momentum. What makes the difference is how the strain-stress relation is formulated for different materials.
>
> To solve advection equations, the symplectic Euler scheme works well for soft materials. To achieve a higher accuracy but still keep the scheme explicit, we can use higher-order schemes such as Runge–Kutta method. However, stiff materials require much smaller time step sizes to maintain stability, making explicit solver inefficient. We plan to explore implicit schemes, such as backward Euler, and develop new backward differentiation formula. We have included it as future work in the revised manuscript.

---

### Official Review · Reviewer_jLz5 · 2022-10-20

**Confidence:** 4
**Correctness:** 4
**Technical Novelty And Significance:** 3
**Empirical Novelty And Significance:** 3
**Recommendation:** 8

**Clarity, Quality, Novelty And Reproducibility:**

Clarity

The paper is well-written, but it's not always easy to follow, and some details are only superficially discussed, although that is understandable in a conference publication. Some details about the implementation are somehow hidden or difficult to find :

- Is the method able to automatically distinguish between Newtonian fluids, non-Newtonian fluids, granular media, deformable solids, and plasticine? Does each material class need to be fixed beforehand since it employs different constitutive equations (strain-stress)? The system would need a material property classifier in a more general application.
- The material parameters seem to be uniform. What could be the adaptation for non-uniform materials? Does the MPM support more complex materials?
- Why is the Oracle baseline so poor in comparison with other methods? Is there any logical explanation for it?.

Quality

The paper is of high quality in general and solves a very difficult problem. The results are well-presented and motivated. The state-of-the-art is well-evaluated, and the experimental evaluation is convincing.

Novelty

This paper combines two existing methods (NERF and a Differentiable Physics Engine) to solve a very complex problem. These methods work with very different representations. The paper proposes a well-grounded strategy to combine them and shows that it works well in photorealistic scenes with large deformations.  It makes the paper novelty more than sufficient.

Reproducibility

Some of the details regarding the MPM method are hidden. It would be very beneficial for the community to publish the code or a more detailed report about the work.




**Strength And Weaknesses:**

Strengths

- The combination of NERF and differentiable continuum mechanics to identify a system's physical parameters and geometry is new.
- The method is well constructed, and each step is properly justified. The change from Eulerian to Lagrangian is a very clever way to combine the physical constraints and the NERF model.
- The results are impressive, even though it only uses synthetically generated scenes. The method is even outperforming the oracle baseline, where the geometry is known.

Weaknesses
- The paper does not include results with real data, which leaves some questions unanswered. (see my comments in the next section)
- The dataset includes only gravity as an external force and a ground plane whose position is known. Other more complex scenarios are not dealt with in the paper.
- The camera poses must be accurate and vary significantly to obtain good results with a NERF method. In practice, this introduces some constraints on the scene, such as a rigid background with a sufficient amount of texture to run structure-from-motion. These constraints are a bit ignored in the paper.

**Summary Of The Paper:**

The paper presents a method to recover a scene's geometry and physical parameters from a collection of images. It combines a neural radiance field method with a differentiable physics engine to jointly recover a deforming object's geometry, color, and physical parameters using continuum mechanics modeling.

The NERF model is dynamic to adapt the geometry over time and is based on the direct voxel grid optimization proposed by Sun et al. (2022) that greatly speeds up training a NERF. The differentiable physics engine is based on the MPM included in DIFFTAICHI.

The paper describes the required steps to convert from the NERF representation of the geometry to particles that move with physical constraints. The NERF and physical parameters are recovered by optimizing a photometric cost function where the physical constraints are implicitly included in the dynamic part of the NERF.

The experiments use synthetically generated scenes with various continuum materials, including Newtonian fluids, non-Newtonian fluids, granular media, deformable solids, and plasticine.



**Summary Of The Review:**

The paper tackles a very difficult problem by combining two recent methods, mixing NERF with continuum mechanics models to identify the physical parameters of a scene from monocular images.

It is well-written and sufficiently clear.

The results are convincing

---

> ### Author Response · Authors · 2022-11-18
> **Response to Reviewer jLz5**
>
> We thank the reviewer for the valuable feedback and positive comments. We address each of the concerns and questions below.
>
> > **Q1:** The paper does not include results with real data.
>
> **A1:** We have included a real-world example on our Demo Website (https://sites.google.com/view/pac-nerf/real-world-example). We used four RealSense cameras to record an elastic ball falling onto the ground. Our reconstructed scene qualitatively matches the targeted videos, which validates that the rendering loss is effectively minimized. We note that due to sparsity of views and imperfect calibrations, the initial geometry is not accurately estimated. Also, the ground is not ideally rigid in the recorded data, which make the reconstructed ball appear to be softer than the ball in the real video. This result is the best we can achieve with our current capture systems. A more sophisticated multi-view capture system should increase the reconstruction quality. We leave it as a future work.
>
> > **Q2:** The dataset includes only gravity as an external force and a ground plane whose position is known. Other more complex scenarios are not dealt with in the paper.
>
> **A2:** Known boundary conditions are trivially to be added in MPM simulations. Two extra experiments with more complex boundary conditions have been included on the Demo Website (https://sites.google.com/view/pac-nerf/complex-bc) and also in the appendix B of the revised manuscript, where an elastic ropes are dropped on two cylinders. PlasticineLab [1] also uses collision objects to manipulate soft bodies, one potential application of our framework would be manipulating NeRF-represented soft objects. We have included these discussions in the revised manuscript.
>
> [1] Huang, Zhihao, et al. “PlasticineLab: A Soft-Body Manipulation Benchmark with Differentiable Physics”. ICLR 2021.
>
> > **Q3:** The camera poses must be accurate and vary significantly to obtain good results with a NERF method. In practice, this introduces some constraints on the scene, such as a rigid background with a sufficient amount of texture to run structure-from-motion. These constraints are a bit ignored in the paper.
>
> **A3:** We agree that this is one of our limitation, as revealed in our real-world experiment. We will have discussed this limitations arising from NeRF in our revised manuscript.
>
>
> > **Q4:** Is the method able to automatically distinguish between Newtonian fluids, non-Newtonian fluids, granular media, deformable solids, and plasticine? Does each material class need to be fixed beforehand since it employs different constitutive equations (strain-stress)? The system would need a material property classifier in a more general application.
>
> **A4:** Our method cannot automatically distinguish between different materials, because such a classification problem is not differentiable. We have listed this limitation in our revision.
>
>
> > **Q5:** The material parameters seem to be uniform. What could be the adaptation for non-uniform materials? Does the MPM support more complex materials?
>
> **A5:** In MPM, physical parameters are defined per-particle, non-uniform materials are supported naturally. We choose to let all particle share the same parameter for simplicity. For non-uniform material, we only need to change the trainable parameters of the MPM simulator to be per-particle physical parameters. As long as the constitutive equation and the plastic return mapping for the material is known, we can use MPM to simulate it.
>
>
> > **Q6:** Why is the Oracle baseline so poor in comparison with other methods? Is there any logical explanation for it?
>
> **A6:** The oracle baseline uses Chamfer distance for supervision. We conjecture the reason that our method can achieve a little better results is that the multi-view rendering loss is less sensitive to design variables than the Chamfer distance. For a query point on a ray, its feature is determined by particles in its one-ring surrounding cells. And the color of that ray is the integral along the ray. However, each oracle particle is only "linked" to its surrounding MPM particles. So the gradient of rendering loss w.r.t one pixel will be backpropagated onto far more particles than the gradient of Chamfer distance w.r.t one oracle particle. This potentially makes the rendering loss smoother than the Chamfer distance and easy to optimize.
>
> > **Q7:** Some of the details regarding the MPM method are hidden. It would be very beneficial for the community to publish the code or a more detailed report about the work.
>
> **A7:** We will open-source our code upon publication.

---

### Official Review · Reviewer_QRFe · 2022-10-20

**Confidence:** 4
**Correctness:** 4
**Technical Novelty And Significance:** 3
**Empirical Novelty And Significance:** 3
**Recommendation:** 8

**Clarity, Quality, Novelty And Reproducibility:**

The paper is overall clear. I believe this work is interesting and should be able to expand the scope of NeRF reconstruction.

**Strength And Weaknesses:**

Strength: Including particle dynamics into NeRF model is novel and it is a very useful to extend NeRF reconstruction beyond static scene/rigid object movements.

Weakness: The proposed method was only evaluated using synthetic data. It would be great if the authors can include some real-world examples as the real-world observation might not strictly follow the particle dynamic system studied in this paper. Yet, I would expect to see some interesting reconstructions based on the proposed model.

**Summary Of The Paper:**

This paper presents a new NeRF model, PAC-NeRF, that incorporate physical particle dynamic into the estimation to simultaneously estimate particle parameters and geometries. The particle dynamics are assumed to follow the conservation laws of continuum mechanics, and a hybrid Eulerian-Lagrangian representation is proposed to solve this dynamic system. The proposed method was trained and evaluated using synthetic data, and with the high quality inputs that satisfied the particle system assumption, it demonstrate better results than the previous dynamic-NeRF (D-NeRF) model.

**Summary Of The Review:**

Please check my comments above. I like this paper and I think this work is of high quality to be accepted.

---

> ### Author Response · Authors · 2022-11-18
> **Response to Reviewers QRFe**
>
> We thank the reviewer for the valuable feedback and positive comments. We address your concern below.
>
> > **Q1:** The proposed method was only evaluated using synthetic data. It would be great if the authors can include some real-world examples as the real-world observation might not strictly follow the particle dynamic system studied in this paper. Yet, I would expect to see some interesting reconstructions based on the proposed model.
>
> **A1:** We agree that real-world data will make our work data more interesting. We have included a real-world example on our Demo Website (https://sites.google.com/view/pac-nerf/real-world-example). We used four RealSense cameras to record an elastic ball falling onto the ground. Our reconstructed scene qualitatively matches the targeted videos, which validates that the rendering loss is effectively minimized. We note that due to sparsity of views and imperfect calibrations, the initial geometry is not accurately estimated. Also, the ground is not ideally rigid in the recorded data, which make the reconstructed ball appear to be softer than the ball in the real video. This result is the best we can achieve with our current capture systems. A more sophisticated multi-view capture system should increase the reconstruction quality. We leave it as a future work.
>
> For the generality of particle dynamic system, we would like to point out that MPM is a PDE-consistent method, just like FEM. As long as the physical procedure can be modeled by a well-defined PDE, MPM can be used to solve the PDE and it will converges to the true solution under refinement (continuously decreasing grid size). MPM can achieve high-accuracy approximation to real-world data [1].
>
> [1] Gaume, Johan, et al. "Dynamic anticrack propagation in snow." Nature communications 9.1 (2018): 1-10.

---

### Official Review · Reviewer_seww · 2022-10-25

**Confidence:** 4
**Correctness:** 4
**Technical Novelty And Significance:** 4
**Empirical Novelty And Significance:** 4
**Recommendation:** 8

**Clarity, Quality, Novelty And Reproducibility:**

This paper is written clearly and easy to understand. The novelties of this paper are significant.

**Strength And Weaknesses:**

Strengths:
1. This motivation of this paper is very clear and the problem studied in this work is of great value to the field of machine vision.
2. The proposed method has strong theoretical support and easy to understand.
3. Extensive experiments are conducted to validate the performance of the proposed method.

Weaknesses:
Since there are no existing approaches that estimate both geometry and material properties from videos, the authors make a best-effort comparison by blending combinations of state-of-the-art approaches for each sub-task. Such comparison seems not sufficient.

**Summary Of The Paper:**

This paper proposes Physics Augmented Continuum Neural Radiance Fields (PAC-NeRF), to estimate both the unknown geometry and physical parameters of highly dynamic objects from multi-view videos. The core idea of this paper is a hybrid Eulerian-Lagrangian representation of the neural radiance field. The proposed method on a wide variety of complex systems outperforms state of the art methods by up to two orders of magnitude.

**Summary Of The Review:**

In summary, this paper studies an interesting problem and propose an effective solution. Experiments demonstrate it effectiveness.

---

> ### Author Response · Authors · 2022-11-18
> **Response to Reviewer seww**
>
> We thank the reviewer for the valuable feedback and positive comments. We address your concern below:
>
> > **Q1:** Since there are no existing approaches that estimate both geometry and material properties from videos, the authors make a best-effort comparison by blending combinations of state-of-the-art approaches for each sub-task. Such comparison seems not sufficient.
>
> **A1:** We have included a real-world example on our Demo Website (https://sites.google.com/view/pac-nerf/real-world-example). We used four RealSense cameras to record an elastic ball falling onto the ground. To extend our comparisons, we tried $\nabla$Sim on this data as well. The experiment results are included in the appendix D of the revised manuscript: the qualitative comparison shows that PAC-NeRF can more effectively minimize rendering loss than $\nabla$Sim.

---

### Public Comment · ~Yunbo_Wang1 · 2022-11-16
**Is it possible to compare the model with another NeRF-based fluid renderer?**

Hi authors,

Interesting paper and solid experiments!

I am working on intuitive physics also, and I note that the general idea of solving the inverse problem by integrating a differentiable dynamics model with an implicit neural renderer is somewhat related to the following work. If possible, could you please compare them for particle estimation, novel view synthesis, and rendering quality at future time steps?

[Guan et al., 2022] NeuroFluid: Fluid dynamics grounding with particle-driven neural radiance fields. ICML 2022.

Thanks!

---

> ### Author Response · Authors · 2022-11-18
> **Response to Yunbo Wang**
>
> Thanks for your comments! The reference above uses intuitive physics to get better predictions, while our work focuses on interpreting the data using continuum physical models. However, we share the same overall idea to combine NeRF with physics to get a version of dynamic NeRF. We have cited your work in our revised manuscript.

---

### Author Response · Authors · 2022-11-18
**General Response**

We sincerely thank all the reviewers for your constructive feedback and encouraging review. In addition to the detailed comments to each reviewer’s questions, we would like to highlight our key contributions, as well as the new experiments we added during the rebuttal phase. The revised parts are marked with red in the manuscript.

## Contributions

We thank the reviewers for acknowledging that

* PAC-NeRF tackles an interesting [seww] and challenging [jLz5] problem.
* PAC-NeRF novelly combines NeRF and physical constraints [QRFe, jLz5, eVdM]. It is an usful extension of NeRF [QRFe].
* PAC-NeRF supports a wide range of different physical systems [seww, jLz5, eVdM]
* PAC-NeRF outperforms state of the art methods [seww, QRFe, eVdM] and the experiment is convincing [seww, jLz5].

We would like to restate this work's novelty and technical contributions:

* PAC-NeRF solves a very challenging untackled task that estimates **both** the geometric structure and physical parameters of a complex dynamic system.
* PAC-NeRF associates the NeRF with the **continuum assumption** so that the appearance can be advected following physical laws strictly. Its hybrid representation seamlessly blends the Eulerian nature of NeRF with MPM’s Lagrangian particle dynamics.
* PAC-NeRF works well on a wide range of highly dynamic systems, including Newtonian/non-Newtonian fluids, elastic bodies, plasticines and sand.

## New Expriments

* The biggest concern among reviewers is the lack of some **real-world experiment** [QRFe, jLz5]. We have included a real-world example on our Demo Website (https://sites.google.com/view/pac-nerf/real-world-example). We used four RealSense cameras to record an elastic ball falling onto the ground. Our reconstructed scene qualitatively matches the targeted videos, which validates that the rendering loss is effectively minimized. We note that due to sparsity of views and imperfect calibrations, the initial geometry is not accurately estimated. Also, the ground is not ideally rigid in the recorded data, which make the reconstructed ball appear to be softer than the ball in the real video. This result is the best we can achieve with our current capture systems. A more sophisticated multi-view capture system should increase the reconstruction quality. We leave it as a future work.
* We also included a **new synthetic scenerio** with a more **complex boundary contion** [jLz5] on our Demo Website (https://sites.google.com/view/pac-nerf/complex-bc): an elastic rope falls onto two rigid cylinders. PAC-NeRF can accurately infer physical parameters.


## Writting
We follows the suggestions of reviews to list our limitations more clearly:
* PAC-NeRF requires accurate camera parameters [jLz5].
* PAC-NeRF assumes the underlying system follows continuum mechanics [EvdM]
* PAC-NeRF cannot automatically change materials types [jLz5].

---

### Decision · Program_Chairs · 2023-01-20

**Decision:**

Accept: notable-top-25%

**Justification For Why Not Higher Score:**

- Limiting assumptions: robustness with respect to camera pose, fixed material class


**Justification For Why Not Lower Score:**

- Very nice combination between NeRF and physics simulators using the Eulerian-Lagrangian formulation
- Pioneer work with a important potential impact

**Metareview: Summary, Strengths And Weaknesses:**

This paper aims at recovering both the geometric structure and the physical properties of objects from multi-view video sequences. The proposed approach combines a neural radiance field (NeRF) with a differentiable physics engine to jointly recover objects' geometry, appearance, and physical parameters. A hybrid Eulerian-Lagrangian formulation is introduced to combine NeRF and differentiable physics so that advecting geometry and appearance. Experiments are conducted on various synthetic videos including a wide range of objects with varying environment lighting conditions and ground textures.
The paper initially received positive reviews, with three accept (8) recommendations, and one borderline accept (6) recommendation. The reviewers pointed out the relevance and novelty of the methods, especially the non-trivial combination of NeRF and differentiable physics with the hybrid Eulerian-Lagrangian formulation. The limitations pointed out by reviewers were related to the absence of evaluation on real data, and requested to specify the assumptions of the methods and its regime of validity. During the rebuttal period, the authors added a new sets of experiments on real data, and provided a discussion regarding the assumption if the approach and its limitations. The paper has been updated accordingly.

The AC carefully reads the submission and discussions. The AC considers that the method is a solid combination between NeRF and differentiable physics, with a particularly neat solution based on the hybrid Eulerian-Lagrangian formulation. The approach pioneers the joint estimation of geometrical and physical properties from image sequences, and can thus have an important impact in the field. Since the experimental comparisons do not show any direct competitors, the added evaluations on real data is important. The discussion on limitations of the approach on the updated paper is also helpful for further developments. The AC thus recommends paper acceptance.

**Note From Pc:**

if the above contains the word "oral" or "spotlight" please see: "oral" presentation means -> notable-top-5% and "spotlight" means -> notable-top-25%. As stated in our emails, we are disassociating presentation type from AC recommendations